# Comparative Assessment of Ethanol and Methanol–Ethanol Blends with Gasoline in SI Engine for Sustainable Development

Muhammad Usman [1,*] , Muhammad Ali Ijaz Malik [2] , Tariq Nawaz Chaudhary [1] , Fahid Riaz [3] ,
Sohaib Raza [1], Muhammad Abubakar [1], Farrukh Ahmad Malik [1], Hafiz Muhammad Ahmad [1], Yasser Fouad [4] ,
Muhammad Mujtaba Abbas [5,*] and Muhammad Abul Kalam [6]

[1] Department of Mechanical Engineering, University of Engineering and Technology, Lahore 54890, Pakistan; tariq.nawaz@uet.edu.pk (T.N.C.)

[2] Department of Mechanical Engineering, Superior University, Raiwind Road, Lahore 54000, Pakistan; muhammadali.ijaz@superior.edu.pk

[3] Mechanical Engineering Department, Abu Dhabi University, Abu Dhabi P.O. Box 59911, United Arab Emirates

[4] Department of Applied Mechanical Engineering, College of Applied Engineering, Muzahimiyah Branch, King Saud University, P.O. Box 800, Riyadh 11421, Saudi Arabia

[5] Department of Mechanical Engineering, University of Engineering and Technology (New Campus), Lahore 54890, Pakistan

[6] School of Civil and Environmental Engineering, FEIT, University of Technology Sydney, Ultimo, NSW 2007, Australia

\* Correspondence: muhammadusman@uet.edu.pk (M.U.); m.mujtaba@uet.edu.pk (M.M.A.)

**Abstract:** Growing environmental concerns over global warming and depleting fossil fuel reserves are compelling researchers to investigate green fuels such as alcoholic fuels that not only show the concrete decrement in emissions but also enhance engine performance. The current study emphasizes the influence of different alcoholic fuel blends in gasoline on engine performance and emissions for an engine speed ranging from 1200 to 4400 rpm. The obtained performance results demonstrate that the brake power and brake thermal efficiency (BTE) increased with an incrementing blend percentage of ethanol and methanol in gasoline (EM). The minimum brake specific fuel consumption (BSFC) was ascertained using pure gasoline followed by E2 and then E5M5. The $NO_x$ and $CO_2$ emissions can be described in the decreasing order of E, EM and gasoline due to same trend of exhaust gas temperature (EGT). CO results were in reverse order of $CO_2$. HC emissions were found in the increasing order of E, EM and pure gasoline. E10 performed better among all blends in terms of less exhaust emissions and engine performance. However, EM blended with gasoline significantly reduced $NO_x$. E5M5 produced 1.9% lower $NO_x$ emission compared to E10 owing to 1.2% lower EGT. Moreover, greenhouse gases such as $CO_2$, which is mainly responsible for global warming reducing by 1.1% in case E5M5 as compared to E10.

**Keywords:** SI engine; ethanol blends; ethanol–methanol blends; engine performance parameters; engine exhaust emissions

## 1. Introduction

The exploration of energy resources and their effective utilization constitute the prime focus of ongoing research. The major means of meeting energy requirements is fossil fuels [1]. The use of fossil fuels is expanding on a daily basis with the growing energy needs of the population [2]. One of the major downsides of these fuels is the fact that their emissions are accompanied by major environmental risks, i.e., global warming [3–5]. The other problem is the depletion in resources of fossil fuels and, as per projections, these resources will be depleted within 40 years [6]. Whilst the major part of today's energy consumption comes from fossil fuels, in the future, it will become very difficult to fulfill our energy needs based on fossil fuels. Considering this situation, the need to use clean energy

is increasing and the world is looking for cleaner fuels to use in automobiles and in other applications [7,8]. The transportation sector is a major consumer of the world's primary energy, representing approximately 18% [4]. Consequently, today's research is focused on the application of alternative and cleaner fuels which will improve engine performance and decrease emissions. Alcoholic fuels are one such alternative to gasoline. Ethanol and methanol comprise those physical properties that make them good alternatives to gasoline to be used in spark ignition (SI) engines [9–11]. Additionally, these fusel oils (ethanol and methanol) do not require any significant modification in the SI engine, which makes them suitable alternative to gasoline. Both of these fuels are considered as among the best alcoholic fuels owing to their higher octane rating, higher flashpoint, higher volumetric efficiency and higher latent heat of vaporization (LHV) [12,13]. The ratio of density of the air–fuel mixture drawn into the engine cylinder at atmospheric pressure during intake stroke to the density of the same volume of air in the intake manifold is known as volumetric efficiency. This directly reflects the fuel consumption, as more volumetric efficiency means lower fuel utilization for generating the same power.

Fuel additives can play a significant role in optimizing engine performance by improving the fuel quality and ensuring that it burns more efficiently. Fuel additives can improve the fuel economy as fuel burns with higher efficiency, increasing engine power output and reducing emissions. Elfasakhany [14] used methanol and ethanol–methanol additives in gasoline in order to investigate the exhaust emissions from an SI engine ran on pure gasoline and gasoline blended with additives. He found that, when added to gasoline, methanol enabled a 6% and 5.5% decline in HC and CO emissions, respectively, in comparison with the ethanol–methanol blend. However, when added to gasoline, ethanol produced 5% and 2% higher CO and HC emissions, respectively, in comparison to ethanol. The ethanol–gasoline fuel showed a decrease of 31% and 14% in CO and HC, respectively, in contrast to gasoline. For EM3 (3 vol.% ethanol and methanol in gasoline), CO and HC decreased by 17% and 10%, respectively; CO and HC decreased by 35% and 15% for EM7, respectively, and decreased by 46% and 23% for EM10 in comparison to gasoline.

Ors et al. [15] executed an experiment on an SI engine by blending methanol and ethanol with gasoline. They found that the addition of methanol increased the brake-specific fuel consumption (BSFC) values by 10.3% in comparison to the addition of ethanol, while the brake thermal efficiency (BTE) was reduced by 6.12%. The addition of methanol decreased the HC, $CO_2$, $NO_x$ and CO emissions by 4.75%, 6.48%, 9.16% and 26.6%, respectively, in comparison to ethanol. Usman et al. [16] ascertained the performance of the SI engine when acetone was added to gasoline. A10 (10% acetone in 90% gasoline) produced 11.74% and 12.05% higher brake power (BP) and BTE, respectively, at a 6.72% lower BSFC. The CO, $CO_2$ and HC emissions declined by 56.54%, 33.67% and 50% in the case of the A10 fuel. However, it was concluded that acetone appeared to be more detrimental to lubricant oil for the existing metallurgy of the engine. In another study, Usman et al. [17] recorded the impact of butanol–gasoline on the SI engine performance. They found 12.15%, 3.25% and 28% higher BP, BTE and exhaust gas temperature (EGT), respectively, in the case of B12 (12% butanol in 88% gasoline). However, the CO and HC emissions declined by 31% and 27% in the case of B12, respectively. They also highlighted that, as the concentration of butanol increased, the lubricant oil started deteriorating early.

Veza et al. [18] performed an experiment on the homogeneous combustion compression ignition (HCCI) engine fueled with an acetone, butanol and ethanol blend with diesel (ABE). They found that the oxygen content in ABE was primarily accountable for the improved combustion which ultimately reduced the CO, $CO_2$, $NO_x$ and HC emissions. However, the soot and PM emissions remain unaffected by the ABE. However, a higher BTE and lower in-cylinder pressure, EGT and BSFC were observed for the ABE blended fuel. Yusuf et al. [19] ran a diesel engine at 1800 rpm under 50% loading conditions fueled with cerium-dioxide ($CeO_2$) nanoparticles in waste cooking oil biodiesel. They found an upsurge in the cylinder pressure, heat release rate (HRR) and $NO_x$ emissions, while the HC and CO emissions were decreased for the blended fuel. Soudagar et al. [20] evaluated

the impact of nanoparticles in diesel and biodiesel blends. They found that the addition of metal-based (manganese, nickel, magnesium, cobalt) additives and oxygenated additives to the biodiesel blend reduced the density, viscosity and flash point due to the increased oxygen content in the blend. These properties are responsible for improved combustion, better fuel spray characteristics, reduced emissions and ultimately, an improved engine performance. They also found that the presence of oxygenated additives (diethyl ether, isobutanol, ethanol) and metallic additives was responsible for reduced BSFC owing to their higher calorific value, catalytic oxidation enhancement and complete combustion of blended fuel.

Sivalakshmi et al. [21] examined the performance of di-ethyl ether (DEE) in neem oil biodiesel in an engine operated at constant 1500 rpm under different loading conditions. They found an increase in $NO_x$ emission, but CO, $CO_2$, HC and smoke emissions were considerably decreased. Akshatha et al. [22] found that the mixing of DEE in neem oil significantly increased the BTE and reduced BSFC. Soudagar et al. [23] used a strontium–zinc oxide additive in biodiesel (20% Ricinus communis and 80% diesel). They found 9.55%, 20.83% and 24.35% increases in in-cylinder pressure, BTE and HRR, respectively, while the ignition delay, BSFC, combustion duration as well as the $CO_2$, CO and HC emissions decreased by 20.64%, 20.07%, 14.5% as well as by 34.9%, 47.63% and 26.81%, respectively, with a minor rise in $NO_x$. In contrast with gasoline, the CO and HC emissions decreased by approximately 17% and 10% using EM3 (3 vol.% ethanol and methanol in gasoline), while they decreased by approximately 35% and 15% or 46% and 23%, respectively, when using EM7 or EM10. Malik et al. [13] concluded with a comparison between M12 (12% methanol in 88% gasoline) and pure gasoline. The results depict the increase in brake power (BP) of 6.69% and a highest achieved efficiency of 23.69% for M12. However, the exhaust gas temperature (EGT), $NO_x$ and $CO_2$ also increased by 16.91%, 27.58 and 2.61% for M12, respectively. The higher temperature was responsible for greater greenhouse gas emissions and the early degradation of the lubricant oil. The kinematic viscosity, total acid number (TAN) and ash content for the lubricant oil operated on M12 increased by 4.57%, 10.23% and 0.97%, respectively.

Mallikarjun et al. [24] observed the influence of 3–15% methanol blends for distinct load settings. They found higher BTE, $CO_2$ and $NO_x$ and decreased HC and CO emissions. Under wide-open throttle (WOT) condition, Prasad et al. [25] performed the experiment on a single-cylinder engine and fueled it with methanol-mixed fuel at sustained 14° BTDC ignition timing at three distinct compression ratios (8, 9 and 10). The combustion efficiency was improved for the methanol-blended fuel when the compression ratio increased, and the HRR and peak pressure also increased by 30% and 27.5%, respectively. BTE increased by 25% at the cost of 19% reduced BSFC. However, the reduction of 30–40% was found for HC, CO and $NO_x$ emissions. Shayan et al. [26] ascertained the exhaust emissions in the case of the methanol-blended gasoline fuel. They kept count of the average drop in HC emissions (24.9%) and CO emissions (23.7%). In contrast to petrol, increases of 7.5% and 17.5% were obtained for $CO_2$ and $NO_x$, respectively. Yontar AA [27] used hydrogen to improve the qualities of gasoline, and after that, he contrasted the outcomes of ethanol and methyl tert butyl ether (MTBE) as well as pure gasoline and hydrogen–gasoline combination with these three fuels. Compared to gasoline, ethanol has a higher BSFC of 29.61%. G98H2 exhibited a BSFC which was approximately 9.24% lower than gasoline. However, ethanol exhibited a BSFC which was approximately 9.24% lower than gasoline. In contrast to gasoline, the ethanol generated 3.11% higher BTE and the G98H2 produced 1.42% higher BTE. For gasoline, MTBE, ethanol and G98H2, the HC formation ranges from 81 to 101 parts per million (ppm), from 80 ppm to 97 ppm and from 77 ppm to 93 ppm, respectively.

Radzali et al. [28] investigated the effects of methanol–gasoline fuel mixes and ambient pressure on exhaust emissions and flame propagation. The flame propagation became wider when the percentage of methanol increased from 0% to 15% in gasoline, which further improved the burn rate. As a consequence of the improved combustion, the $CO_2$ emissions were increased, while the CO, $NO_2$ and HC emissions were lowered. Balki, Sayin and Canakci [29] presented the comparative research between methanol and gasoline on a

196 cc gasoline engine. The achieved results exposed that the torque would rise by 4.7% and that the BSFC would rise by 84% with reference to gasoline.

Malik et al. [1] also examined the behavior of M6 (6% methanol in 94% gasoline) on the SI engine performance. The obtained results revealed 3.72% and 1.38% higher brake power and BTE in the case of M6. However, M6 showed a 1.37% smaller brake specific energy consumption (BSEC) owing to the naturally lower calorific value of methanol. The lower calorific value is responsible for augmented fuel consumption and consequently, the combustion temperature rises. The EGT and NO emissions for M6 were 10.66% and 9.70% higher for the M6 fuel blend, respectively. The kinematic viscosity, total acid number (TAN), flashpoint and ash content of the lubricant oil operated on M6 were increased by 1.28%, 6.03%, 3.49% and 0.48%, respectively, in contrast to lubricant oil operated on gasoline. Ahmed et al. [30] conducted experiments on various methanol blends with gasoline ranging from 0% to 18%. They found that the EGT was 3.38%, 11.43%, 19.48%, 24.65%, 25.53% and 27.38% higher for M3–M8 with an interval of 3% by volume methanol, respectively, whilst $NO_x$ emissions increased by 5.49%, 12.54%, 19.19%, 28.73%, 35.81% and 41.13%, respectively. However, the HC emissions decreased by 2.12%, 4.97%, 7.92%, 11.05%, 14.14% and 17.04% for M3–M8. Thakur et al. [31] deduced that BP increased by 2.31%, 2.77% and 4.16% for E5, E10 and E20, respectively, while maximum increases of 3.5%, 2.5% and 6% in BTE were found for E20, E10 and E40, respectively. Hsieh et al. [32] found that CO and HC emissions declined by 10–90% and 20–80%, respectively, while $CO_2$ emissions increased by 5–25% depending on ethanol–gasoline blend ratio, varying from 5% to 30%. Masum et al. [33] stated that a higher flame speed of methanol helped complete the combustion process, and as a consequence, a higher $NO_x$ emission was achieved for the ethanol–gasoline fuel blend.

The literature review showed that the performance characteristics of the gasoline engine, on average, increased using alcoholic fuels such as ethanol, methanol, etc. The brake power and torque are enhanced using ethanol blended with gasoline. On average, the maximum increase of 20% in the performance parameters was achieved with the ethanol blend, and then, these parameters started to decrease after attaining maximum values [34,35]. Similarly, the performance parameters were also found to be higher for methanol-blended gasoline compared to pure gasoline [36]. The brake power was noticed to be higher for ethanol compared with methanol [37]. The disadvantage of using the ethanol–gasoline blended fuel is that the BSFC increases with increasing concentrations of ethanol [38]. The blend of methanol in gasoline also showed an increase in BSFC [39,40].

The assessment of ethanol and methanol blended with gasoline demonstrated that the BSFC for ethanol was lower than that for methanol [41]. By employing alcoholic fuels, an improvement was observed in the BTE of the gasoline engine and it increased with increasing concentrations [42–44]. When the BTEs of both ethanol and methanol were compared, that of ethanol was higher due to its better combustion compared to methanol [45]. An imperative consideration to adjudicate the combustion completion is the exhaust gas temperature (EGT). Previous findings have shown that the addition of ethanol increased the exhaust temperature in comparison to gasoline, further resulting in more oxides of nitrogen ($NO_x$) emissions [46,47]. Similar results were obtained with methanol [26,47]. The comparison of the both alcoholic fuels showed a higher EGT for ethanol [37]. Using alcoholic fuels such as ethanol, methanol, butanol, etc. with gasoline considerably reduced the emissions [48]. The application of ethanol-blended gasoline also caused the emissions of unburnt hydrocarbon (HC) to drop significantly. It was also found that the emissions go on depleting with increasing concentrations of ethanol, which is due to the better combustion of ethanol-blended gasoline (E) [42,43,49]. The same results were also found for methanol [36]. Methanol generated greater HC contents when compared with ethanol [37]. Carbon monoxide (CO) emissions were observed to reduce using the ethanol blend with gasoline [50].

The use of methanol with gasoline revealed that the CO emissions decreased with increased proportions of methanol [37]. The CO emissions for ethanol were found to be lower

than those for methanol [51]. The results showed a higher carbon dioxide ($CO_2$) content in the exhaust for ethanol and methanol compared with pure gasoline [50,52]. Comparing both of these alcoholic fuels, ethanol showed more $CO_2$ emissions than methanol [37].

Nitrogen oxide ($NO_x$) emissions are related to the exhaust temperature and increase as the exhaust temperature increases. $NO_x$ emissions are greater for ethanol–gasoline blends than they are for pure gasoline [47]. The comparison of methanol blended with gasoline alone also showed an upsurge in the $NO_x$ emissions in contrast with pure gasoline. The results also show that the $NO_x$ emissions were found to be higher for ethanol compared with methanol [37]. Tibaquira et al. opted for the eco-indicator-99 methodology to quantify the impact of the ethanol–gasoline blend on the environment. They obtained the results which declared that the effects on healths of humans, the environment and natural resources were decreased by 1.3%, 1.4% and 12.9%, respectively, when 20% ethanol in 80% gasoline (E20) was used [53].

The studied literature reveals that there is an existing gap in analyzing the impact of the ethanol–methanol blend (EM) in gasoline on exhaust emissions and the performance characteristics of gasoline engine. In this very context, this study is focused on ascertaining the impact of ethanol–methanol as well as those of ethanol blends in gasoline on the performance and exhaust pollutants of a four-stroke SI engine. Additionally, the literature review reveals that $CO_2$ and $NO_x$ emissions increase for alcoholic fuels. $CO_2$ is major a contributor to the greenhouse effect and global warming, while $NO_x$ is an indicator of higher EGT. A higher temperature will evidence premature lubricant oil degradation. Therefore, attempts were made to develop such blends which not only reduced $CO_2$ but also $NO_x$ and EGT in order to maintain optimum conditions for engine performance. This will surely be a valuable addition to research on alcoholic blends. In the current study, it was found that, despite their greater consumption due to their lower calorific value alcoholic fuels, these produce more brake power, EGT, $CO_2$ and $NO_x$ emissions. The higher $CO_2$, EGT and $NO_x$ emissions indicate better fuel combustion and ultimately improve the BTE and decrease the CO and HC emissions.

## 2. Materials and Methods

For the current investigation, ethanol and methanol were procured from Merck Chemicals (Darmstadt, Germany), however, gasoline was procured from Pakistan State Oil (PSO) (Karachi, Pakistan). These were mixed in different proportions, as mentioned in detail in the following sections, in order to ascertain the engine performance and record the exhaust emissions. The lubricant oil SAE 10W30—commercially available as CALTEX Havoline oil—was filled into the engine in order to ascertain the deterioration in lubricant oil after 120 h of engine operation. Table 1 specifies the ascertained lubricant oil parameters along with the test standards and equipment. A spark ignition (SI) engine is a type of internal combustion engine that uses a spark plug to ignite the air–fuel mixture in the combustion chamber at the end of the compression stroke. Therefore, the air–fuel mixture undergoes combustion and expanding gases cause the piston to move downward and turn the crankshaft in order to rotate the wheels of the vehicle. The SI engine uses gasoline as fuel. An SI engine was selected for the current study due to its multiple advantages over diesel engine, including lower emissions, light weight, less noise and vibration as well as fuel efficiency. Figure 1 displays the actual test bed and engine for the evaluation of engine performance.

**Table 1.** Lubricant oil characteristics.

| Lubricant Characteristics | Units | ASTM Standards | Testing Equipment |
| --- | --- | --- | --- |
| Kinematic viscosity at 40 °C | cst | ASTM D-445 | SETA KV-6 viscosity bath |
| Kinematic viscosity at 100 °C | cst | ASTM D-445 | Tamson TV 4000 viscosity bath |
| Total base number | mg KOH/g | ASTM D-2896 | Metrohm titrono plus 877 |
| Flash point | °C | ASTM D-92 | SETA semi-automatic flash point unit |

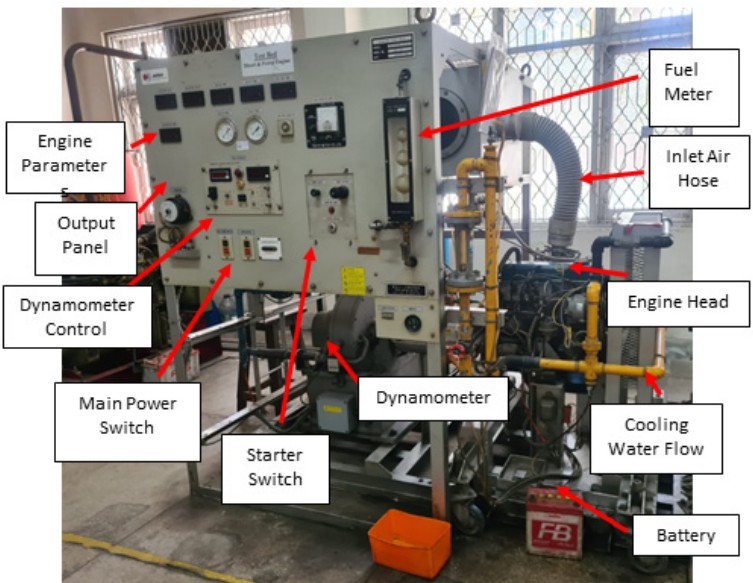

**Figure 1.** Actual test bed and engine.

*2.1. Test Setup*

A water cooled 4-stroke and 4-cylinder gasoline engine modeled as A15 and manufactured by Tokyo Meter Co., Ltd. (Tokyo, Japan), equipped with a test bed of model GWE-80, manufactured by Tokyo Meter Co., Ltd., was used for the experiments. The specific engine was used in the experiment due to its availability in the Thermal Power Systems Laboratory of the University of Engineering and Technology, Lahore. The engine was fitted with a cell motor for the initial start. Further specifications of the test engine are shown in Table 2.

**Table 2.** Technical specifications of the SI engine [a].

| Parameter | Value |
|---|---|
| Bore (cm) | 7.6 |
| Displacement ($cm^3$) | 1487 |
| Stroke length (cm) | 8.2 |
| Compression ratio | 9.0 |
| Link ratio (l/r) | 3.24 |
| Maximum torque (Nm) | 116.6 |
| Power rating (kW/rpm) | 57.4/5600 |

[a] Source: Tokyo Meter Co., Ltd.

A magnetic stirrer (hot plate 78-1) was used for homogeneous mixing of ethanol, methanol and gasoline for 25 min. The fuel blends were then filled into containers immediately after their preparation to avoid phase separation and ensure homogeneity. The fuel flow meter with three burettes of 30 mL, 50 mL and 100 mL installed on the engine test bed control panel was used for fuel measuring purposes. Time was noted using a digital stopwatch. The time for one burette to be consumed was measured to calculate the fuel consumption and specific fuel consumption. The fuel and air intake mechanism into the engine was controlled by the carburetor. Engine test bed layout plan is presented in Figure 2.

Test bed had an electrodynamometer for exerting brake load on the engine. Therefore, the engine speed and brake power were measured through dynamometer against specific load. For engine speed measurement, an electromagnetic detector tachometer was installed with the test-engine shaft. Engine brake power, as a function of engine speed and load on the brake applied, was calculated. The EMS 5002 exhaust gas analyzer was used to analyze the exhaust gases of the engine.

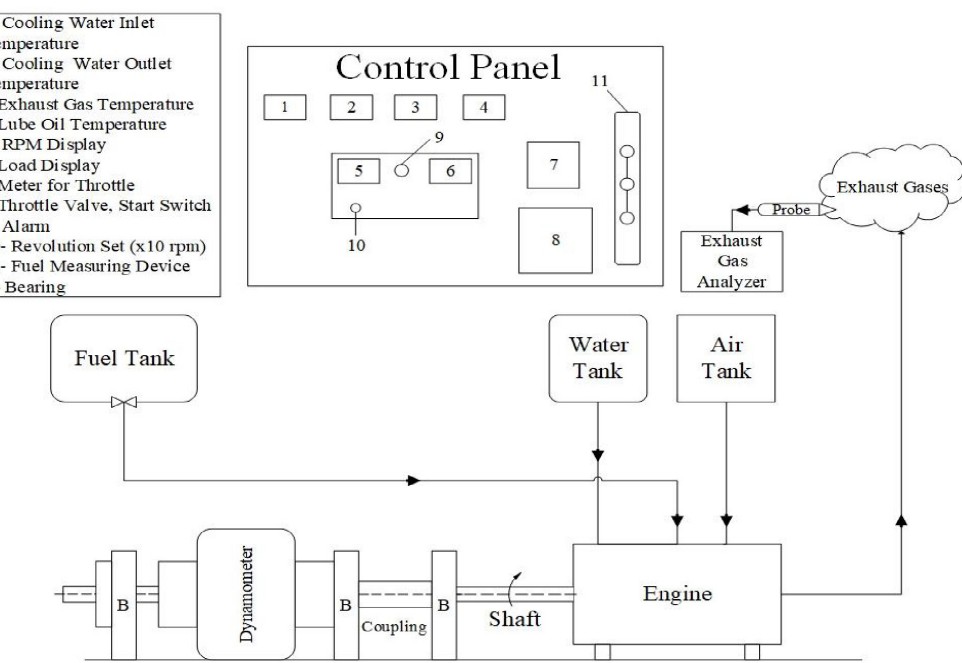

**Figure 2.** Engine test bed layout plan.

*2.2. Uncertainty Analysis*

The extent of accuracy of the measured parameters can be ascertained through uncertainty analysis. This also gives the extent of error in each measurement of the experimental setup. The measurable range, accuracy and uncertainty of the parameters in the measurements are given in Table 3. The overall uncertainty of the experimental setup ($U_{exp}$) was evaluated with the help of following equation [32]:

$$U_{exp} = [(U_{HC})^2 + (U_{Speed})^2 + (U_{NOx})^2 + (U_{Power})^2 + (U_{CO})^2 + (U_{CO2})^2 + (U_{BSFC})^2 + (U_{Temp})^2]^{1/2}$$
$$U_{exp} = [(1)^2 + (0.5)^2 + (1)^2 + (1)^2 + (1)^2 + (1)^2 + (0.5)^2 + (0.1)^2]^{\frac{1}{2}}$$
$$U_{exp} = 2.12\%$$

**Table 3.** Measurable range, accuracy and uncertainty in the measurements.

| Parameters | Range Measurable | Accuracy | Uncertainty (%) |
|---|---|---|---|
| HC [a] | 0–2000 ppm | 1 ppm | ±1 |
| Speed [b] | 0–7000 rpm | 2 rpm | ±0.5 |
| NOx [a] | 0–5000 ppm | 1 ppm | ±1 |
| Power [b] | 0–110 kW | 0.5 kW | ±1 |
| CO [a] | 0–10% | 0.01% | ±1 |
| CO2 [a] | 0–20% | 0.1% | ±1 |
| BSFC [c] | - | 0.1 kg/kWh | ±0.5 |
| Temperature [c] | 0–1000 °C | 1 °C | ±0.1 |

Source: [a] EMS-5002 operations manual by Emissions System, Inc. (Whitby, ON, Canada) [b] Electro-dynamometer operations manual by Tokyo Meter Co., Ltd., [c] Engine operations manual by Tokyo Meter Co., Ltd.

*2.3. Test Scheme*

For the specified engine, a chain of experiments was conducted first with pure gasoline and then using different compositions of fuel blends of gasoline with ethanol (E) and a mixture of gasoline, ethanol and methanol (EM). Fuel blends were made as detailed in Table 4, where numbers show the percentages of the alcoholic fuel mixed with gasoline by volume. The characteristics of the gasoline, ethanol and methanol used as fuel are given in Table 5.

**Table 4.** Fuel blends mixing percentages.

| Fuel | Blend Percentage | Fuel | Blend Percentage |
|---|---|---|---|
| E | % ethanol in gasoline | EM | % ethanol and % methanol in gasoline |
| E0 | pure/100% gasoline | E0M0 | pure/100% gasoline |
| E2 | 2% ethanol and 98% gasoline | E1M1 | 1% ethanol and 1% methanol with 98% gasoline |
| E4 | 4% ethanol and 96% gasoline | E2M2 | 2% ethanol and 2% methanol with 96% gasoline |
| E6 | 6% ethanol and 94% gasoline | E3M3 | 3% ethanol and 3% methanol with 94% gasoline |
| E8 | 8% ethanol and 92% gasoline | E4M4 | 4% ethanol and 4% methanol with 92% gasoline |
| E10 | 10% ethanol and 90% gasoline | E5M5 | 5% ethanol and 5% methanol with 90% gasoline |

**Table 5.** Properties of fuels used.

| Properties | Gasoline [a] | Ethanol [b] | Methanol [b] |
|---|---|---|---|
| Phase | Liquid | Liquid | Liquid |
| Molecular formula | $C_8H_{18}$ | $C_2H_5OH$ | $CH_3OH$ |
| Air to fuel ratio (A/F) | 14.7 | 9 | 6.4 |
| Molecular weight (g/mol) | 114 | 46 | 32 |
| Density at 25 °C (g/mL) | 0.736 | 0.785 | 0.791 |
| Calorific value (MJ/kg) | 46 | 26.9 | 20 |
| Boiling point (°C) | 195 | 78 | 65 |
| Octane number | 97 | 108 | 111 |
| Latent heat of vaporization (kJ/kg) | 300 | 846 | 1110 |
| Auto-ignition temperature (°C) | 257 | 425 | 465 |
| Flame velocity (cm/s) | 33 | 50 | 52.3 |
| Oxygen content (%$v/v$) | 0 | 34.73 | 50 |
| Viscosity (mPa · s) | 0.602 | 1.17 | 0.594 |

Source: [a] Pakistan State Oil (PSO); [b] Merck Chemicals.

All experiments were conducted under 60% throttle set conditions and the tests were conducted in the range of 1200–4400 rpm. The fuel consumption was recorded by observing the time required for 30 mL of consumed fuel. At each value of fuel composition and rpm, calculations were carried out for brake power (kW), BSFC as kg/kWh of the output and BTE (%). Heat soaking was kept under observation for accurate values and the engine was given enough time for stable conditions. Values were observed for one minute after the stable conditions of engine performance were attained. Three values, after each 30 s, were taken for each variable to take the average. Taking the average of the variable minimized the error and outputs were reasonable. The zeroing of the exhaust gas analyzer, performed before each observation, was recorded after one minute of stable values for the emissions.

## 3. Results

### 3.1. Comparative Assessment of Engine Performance

Brake power is a measure of the engine power output or the amount of power delivered to the engine shaft after accounting for losses and expressed in kW. The difference in brake power (BP) for various alcoholic fuel blends is described in Figure 3. BP was observed to be higher for E than EM and the least for pure gasoline. This showed that BP increased by increasing the blend percentage of both ethanol and methanol in gasoline. Brake power was noted to be at its maximum for E10, i.e., 23.5 kW, followed by 23.1 kW for E5M5 and at its minimum for pure gasoline (21.6 kW). The maximum values of E10 and E5M5 occurred at 3200 rpm. The brake power augmented with an increment in the concentration of ethanol and methanol in the gasoline. However, the experiment will not be performed on their higher percentages in order to fall in the safe region of engine operations. It was studied in the literature that blends of approximately 10% proportion can be used without any engine modification and produce the most optimized results. It was for this reason that these specific blend percentages were selected. The maximum increases of 5.70% and 4.91% in brake power were achieved for E10 and E5M5, respectively. The higher increase in the ethanol fuel blend than in the ethanol–methanol fuel blend can be explained by the higher calorific value of ethanol. Moreover, the increase in brake power for E blends compared with pure gasoline is accredited to the higher heat of vaporization for ethanol which results in the cooling of the air–fuel charge inside the cylinder, further causing higher charge density. The high density of air–fuel charge puts more pressure on the piston surface, due to which the BP improves [34]. The same is the reason for the case of EM blends [36]. The

higher value of LHV for ethanol compared with methanol has been reasoned with more brake power for E than EM blends [37]. Elfasakhany [54] obtained 5.50%, 0.63%, 1.31% and 2.86% increases in brake power for E10, E3M3, M7M7 and E10M10, respectively. Therefore, the results of the current study can be validated through previous research results.

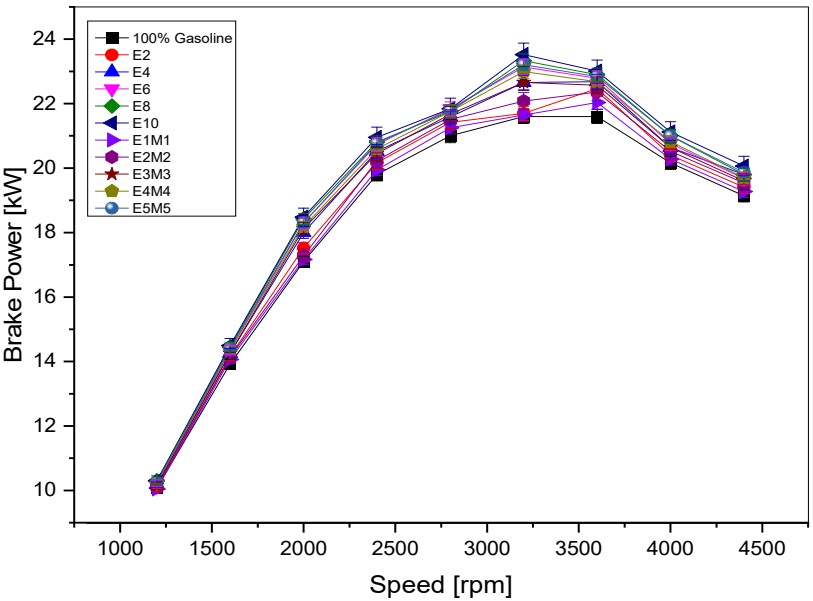

**Figure 3.** Brake power for the E and EM blends.

The naturally enhanced oxygen content and octane rating are mainly responsible for higher BP. The antiknock attributes induced into fuel owing to higher octane rating play a significant role in reducing friction and eventually improving the brake power [25]. Moreover, the quicker propagation of the laminar flame and augmented oxygen content of alcoholic fuels compared to gasoline facilitates the completion of the combustion process before any significant loss from the chamber surface can occur [39].

BSFC is the ratio of the fuel consumption per unit power and depicts the effectiveness of the engine for fuel consumption, expressed as kg/kWh. The trend of BSFC for varying blend ratios and blend types is shown in Figure 4. BSFC first decreases and then increases with the increase in engine speed. This can be credited to greater fuel consumption in the beginning in order to overcome inertial effects and enable the engine's operating conditions. Moreover, thermal losses across engine walls were initially higher for a lower rpm rate, and as a consequence, leads to a higher fuel consumption to reimburse the thermal losses. BSFC steadily declines with the increasing engine rpm, and then, after 2800 rpm, the BSFC starts to upsurge. The combustion undergoes close to stoichiometric conditions during the lowest BSFC conditions. The higher BSFC at the higher engine rpm is because of the higher fuel consumption in order to match the higher power requirements. It is evident that BSFC for E and EM is more than pure gasoline. The minimum BSFC obtained was 0.3451 kg/kWh for pure gasoline followed by 0.3453 kg/kWh for E2 and then 0.34578 kg/kWh for E1M1. The higher values of BSFC for the E blends is due to the fact that ethanol has a lower heating value than gasoline, and in order to produce the same/more power than gasoline, it requires more fuel and hence the BSFC is increased [38]. Likewise, EM blends require more fuel to generate the same power as gasoline [40]. Comparing BSFC for E and EM blends, EM blends produced a higher BSFC owing to the lower calorific value of methanol in comparison to ethanol [55]. In the current study, BSFC increased by 3.67% and 1.97% for E5M5 and E10, respectively, compared to pure gasoline. Kamil and Nazal [56] found that decreases in BSFC of 10%, 14% and 18.4% in the cases of E6M6, M12 and E12, respectively.

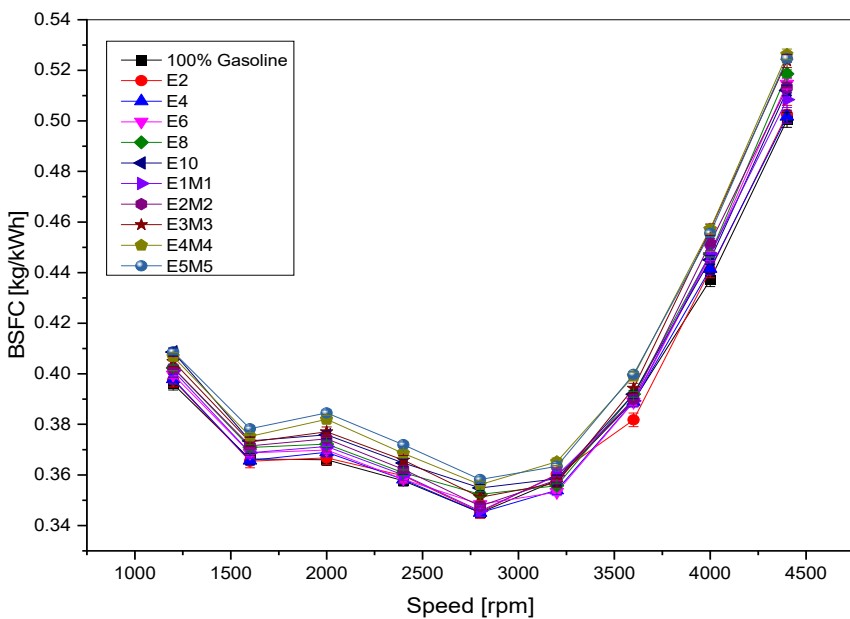

**Figure 4.** BSFC for the E and EM blends.

BTE is a measure how efficiently the engine converts the fuel's heat energy into useful mechanical work, which is expressed as a percentage. BTE represents the amount of engine BP production with respect to fuel energy. A generic BTE trend showed by fuel blends as revealed in Figure 5 discloses the rise in BTE first to a maximum value, and then, eventually BTE falls. It is stated that the engine efficiency is inversely proportional to heat loss. Furthermore, BTE exhibits an inverse relation to the BSFC and calorific value [55]. The higher LHV of alcoholic fuels (ethanol and methanol) is responsible for heat absorption from cylinder walls. As a consequence, the fuel mixture must be compressed to lower and eventually augment the thermal efficiency [57]. The quicker proliferation of the laminar flame for alcoholic fuels marks the faster combustion heat release phenomenon before any significant thermal losses occur and with a better isometric impact [58]. It is clear that BTE decreases in the order of E, EM and pure gasoline. E10 showed the maximum value of BTE, i.e., 23.1%, followed by 22.9% for E5M5 and 21.9% for pure gasoline. The maximum values for the EM and E blends were noticed at 3200 rpm and for pure gasoline at 2800 rpm. Adding ethanol/methanol to gasoline improved the efficiency and the optimum point was shifted to a higher rpm. The higher values of BTE for E and EM blends compared to pure gasoline can be explained by the higher oxygen content in ethanol which improves the combustion [42,50]. Amongst the E and EM blends, E showed higher values of BTE due to the improved combustion and higher lower heating value of ethanol [45]. The antiknock attributes induced into fuel owing to the higher octane rating play a significant role in reducing the friction, which eventually improves the brake output and ultimately increases the BTE [25]. The BTE for E10 and E5M5 increased by 2.35% and 1.53%, respectively, in comparison with pure gasoline. Kamil and Nazal [56] found an increase in the BTE of 23%, 32% and 17% in the cases of E6M6, M12 and E12, respectively. The brake thermal efficiency increased in a lower proportion compared to the increase in BTE, as observed in previous research.

Figure 6 shows the relationship between EGT and the blend percentage of ethanol/ methanol. EGT was observed to be higher for E than for EM and at its minimum for pure gasoline. E10, E5M5 and pure gasoline showed maximum temperatures of 149 °C, 147 °C and 130 °C, respectively. EGT facilitates the interpretation of the combustion quality and the development of exhaust emissions [15]. The literature depicts increasing or decreasing EGT behaviors for alcoholic fuels. It either increases due to the enhanced oxygen content of alcoholic fuels [30], or decreases due to the higher LHV of alcoholic fuels [59,60]. The higher EGT values for E and EM blends in comparison to gasoline can be supported by the reason that the addition of ethanol to gasoline improves the combustion [26,47]. Comparing

values of EGT for E and EM blends showed that E has higher values. As the ethanol and methanol blend percentages increase, the combustion becomes more stoichiometric and higher temperatures are obtained. Ethanol carries a lower heat of vaporization value than methanol, which causes a lesser cooling effect of the air–fuel charge inside the cylinder and a greater in-cylinder temperature. Eventually, the higher EGT for E compared to EM blends can be obtained [37]. The higher EGT for alcoholic fuels due to a lower calorific value was as a consequence accountable for the increased fuel supply into the cylinder. Moreover, the higher oxygen content of the alcoholic blended fuels eventually accounts a better combustion rate due to effective fuel burning. The EGT increases with the increase in engine rpm due to greater fuel burning in order to meet higher power needs. In the current study, EGT for E10 and E5M5 increased by 14.61% and 13.08%, respectively, in comparison with pure gasoline. Kamil and Nazal [56] found that the EGT declined by 1.97%, 3.3% and 1.7% for E6M6, M12 and E12, respectively. The higher EGT in the current study might be due to the higher fuel consumption and combustion in the engine cylinder in order to generate comparable power.

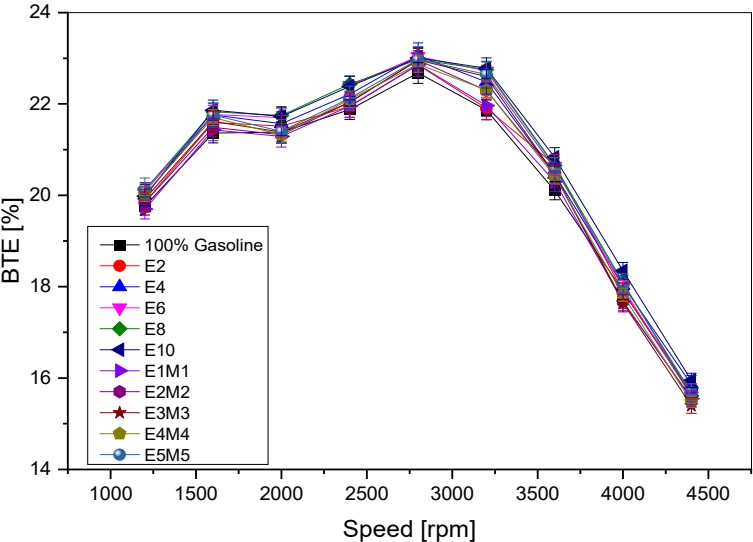

**Figure 5.** BTE for the E and EM blends.

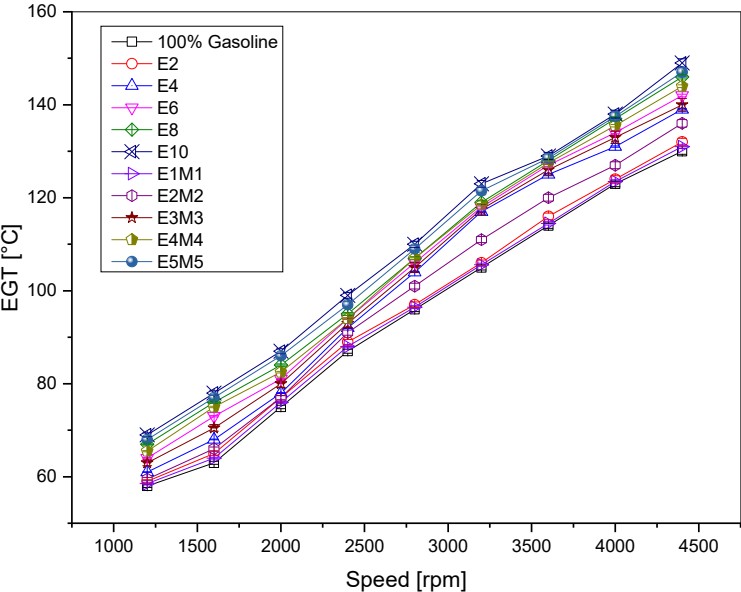

**Figure 6.** EGT for the E and EM blends.

### 3.2. Comparative Assessment of Engine Emissions

The difference in hydro-carbon (HC) concentrations across all the test fuels is displayed in Figure 7. E10 produced the least HC emissions, i.e., 196 ppm compared with 197 ppm and 230 ppm for E5M5 and pure gasoline, respectively. The E and EM blends showed lower values than pure gasoline due to the improved combustion and lower carbon content compared with pure gasoline [36,43]. On average, E10 and E5M5 produced 13.2% and 12.1% lower HC emissions than gasoline. Comparing E and EM blends, E emitted less emissions, which can be credited to its better combustion and lower carbon-to-hydrogen ratio [1,37]. Hydro-carbon emissions result from inappropriate combustion and unburned fuel evaporation into the environment. Moreover, leakage from an exhaust manifold, the fuel state during warmup and cold start, the accretion of incombustible fuel in cracks and oil film deposition are significant factors that contribute to HC emissions [61]. The blending of conventional fuel with oxygenated fuel results in the oxidation of conventional fuel during post flame conditions which can be accounted for as the prime reason for lowering HC emissions [62]. The oxygen content inside alcohols is responsible for effective combustion as oxygen undergoes a reaction with hydrogen and carbon to produce $H_2O$ and $CO_2$, respectively [15]. Consequently, there would be less of a chance of reaction between carbon and hydrogen and HC emissions would be lowered. The naturally higher octane number of alcoholic fuels induced antiknock attributes into fuel which primarily reduce friction, eventually improve combustion, and ultimately decrease HC emission [25]. Elfasakhany [54] obtained 9.78%, 6.72%, 9.26% and 14.57% decreases in HC emissions for E10, E3M3, M7M7 and E10M10, respectively. Therefore, the results of the current study are comparable with previous research results. In the current study, the HC emission decreased by 13.2% for E10 and previous results indicated a decline of 9.78% for E10. The decrease in HC emissions for E5M5 was 12.1%, but previous results showed decreases of 6.72% and 9.26% for E3M3 and E7M7, respectively. This means that HC emissions decline more in the current study.

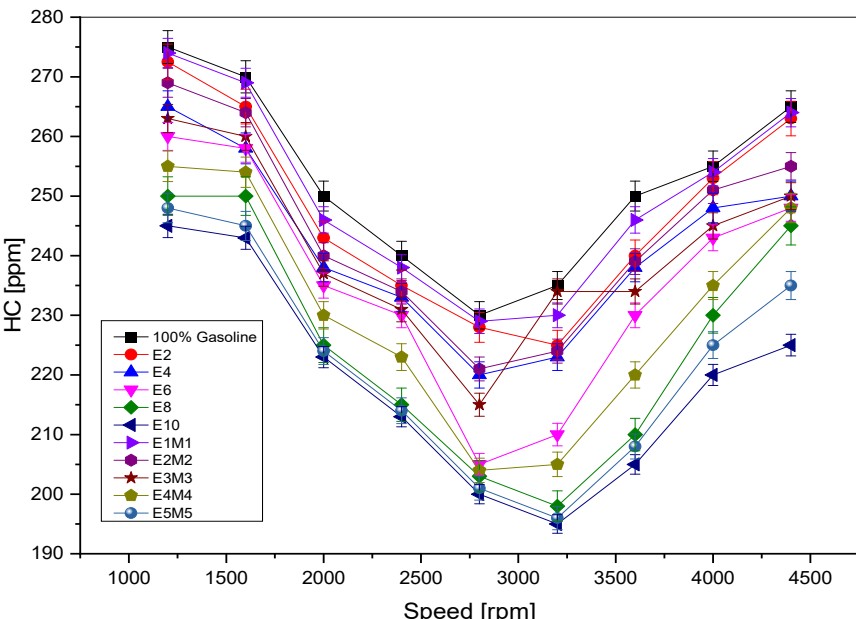

**Figure 7.** HC emissions for the E and EM blends.

The addition of ethanol and methanol decreased CO emissions, as evidenced in Figure 8. The reductions in CO emissions of 36.2% and 32.8% were recorded for E10 and E5M5, respectively. As the percentage of ethanol increased, CO emissions reduced in comparison to gasoline. The reason is that the addition of ethanol in gasoline enhances the oxygen contents, which improves the combustion. The improvement in combustion reduces CO emissions [50]. Lower CO emissions for alcoholic fuel blends can be credited to

a higher oxygen content which increases the oxygen (O)-to-carbon (C) ratio in the cylinder, making it more likely to form carbon dioxide ($CO_2$) instead of CO due to the higher oxygen availability. Similarly, the EM blend gave the least CO emissions compared with pure gasoline. The reduced carbon content in both alcohols also reduced the CO emissions [37]. The EM blend gave greater emissions compared with E due to the increased carbon content when both ethanol and methanol were added to gasoline compared with the addition of only ethanol [51]. The higher LHV of alcoholic fuels accounts for the increased volumetric efficiency (VE), as it guarantees homogeneous mixing due to its better molecular diffusivity and higher flammability limit, ultimately improving the combustion for alcoholic fuels in the engine [63]. Elfasakhany [54] obtained 20.49%, 3.08%, 16.02% and 21.42% decreases in CO emissions for E10, E3M3, M7M7 and E10M10, respectively. Therefore, the results of the current study are comparable with those of previous research results. In the current study, CO emissions decreased by 36.2% for E10 and previous results indicate a decline of 20.49% for E10. The decrease in CO emission for E5M5 was 32.8%, however, previous results depict decreases of 3.08% and 16.02% for E3M3 and E7M7, respectively. This means that CO emissions undergo greater declines in the current study.

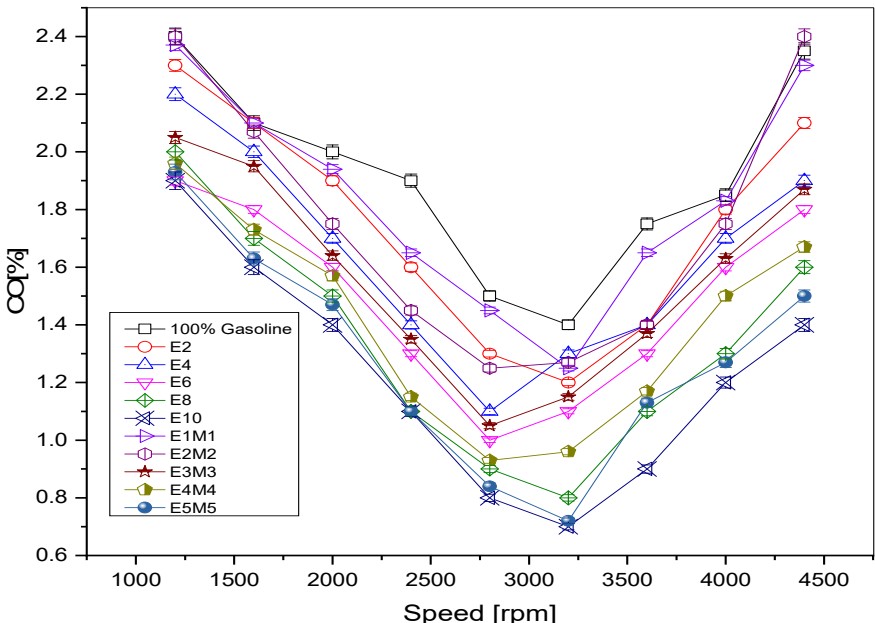

**Figure 8.** CO emissions for the E and EM blends.

E blends showed the maximum carbon dioxide emissions and the EM blends showed the minimum $CO_2$ emissions for pure gasoline (see Figure 9). The $CO_2$ emissions of E blends first increase and reach maximum values, and then finally decrease after reaching the peak. $CO_2$ emissions declined at higher rpm after the optimal speed range due to the inadequate amount of time being available for mixing between fuel and air, finally leading to a decrease in BTE. The greater value of $CO_2$ emissions is attributed to the improved combustion of fuel, leaving behind the less unoxidized CO as can also be verified from the CO trends. The results declare that the percentage of $CO_2$ increased with the increase in rpm, for each fuel used, up to a limit before decreasing. The maximum value of $CO_2$ was observed at 8.15% by E10 followed by 8% for E5M5 and 6.8% for pure gasoline. The maximum values of $CO_2$ for the E and EM blends were observed at 3200 rpm, as compared to gasoline, whose maximum value occurred at 2800 rpm. Blending alcoholic fuels with gasoline shifted the optimum point ahead from 2800 rpm to 3200 rpm. This provided the benefit of allowing the engine to run at higher speed with maximum efficiency, as compared to gasoline. At the optimum point, the combustion process is better; hence, converting the maximum of CO into $CO_2$ through oxidization [50]. $CO_2$ was also observed to increase, almost linearly, with the percentage of the E blend up to E10 and greater than gasoline

values [52]. While comparing the $CO_2$ for E and EM, it was observed that E blends emitted a higher $CO_2$ in exhaust gases as compared to EM. A greater oxygen content in ethanol provided sufficient oxygen for the better combustion in E than in EM [37]. The higher laminar flame speed in the case of alcoholic fuels was responsible for the early completion of combustion before any significant losses from cylinder walls [13]. Both methanol and ethanol comprised carbon (C), hydrogen (H) and oxygen (O), such that C and H underwent a reaction with O during the fuel burning in order to yield $CO_2$ and $H_2O$. Both $CO_2$ and $H_2O$ are the ideal products of combustion [47]. Therefore, $CO_2$ directly linked with BTE. For effective fuel burning, the $CO_2$ emission would be higher. On the other hand, $CO_2$ would be lower for inappropriate fuel burning but subsequently higher for CO emissions. The oxygenated alcohols foster lean burning and optimize combustion by converting CO into $CO_2$ [32,55]. Moreover, the higher $CO_2$ emission produced in the case of alcoholic fuels (ethanol and methanol) can be accredited to a certainly higher oxygen proportion and octane rating. Therefore, antiknock attributes are induced into fuel owing to the octane rating. Consequently, combustion improved at the cost of lower friction and an increase in $CO_2$ emissions [25]. Elfasakhany [54] obtained 6.15%, 2.40%, 5.29% and 6.83% increases in $CO_2$ emissions for E10, E3M3, M7M7 and E10M10, respectively. Therefore, the results of the current study are comparable with previous research results. In the current study, the $CO_2$ emission increased by 13.6% for E10 and previous results indicate an increase of 6.15% for E10. The increment in CO emission for E5M5 was 12.5%, however, the previous results depict decreases of 2.40% and 5.29% for E3M3 and E7M7. This indicates that the $CO_2$ emissions increase more than in the current study.

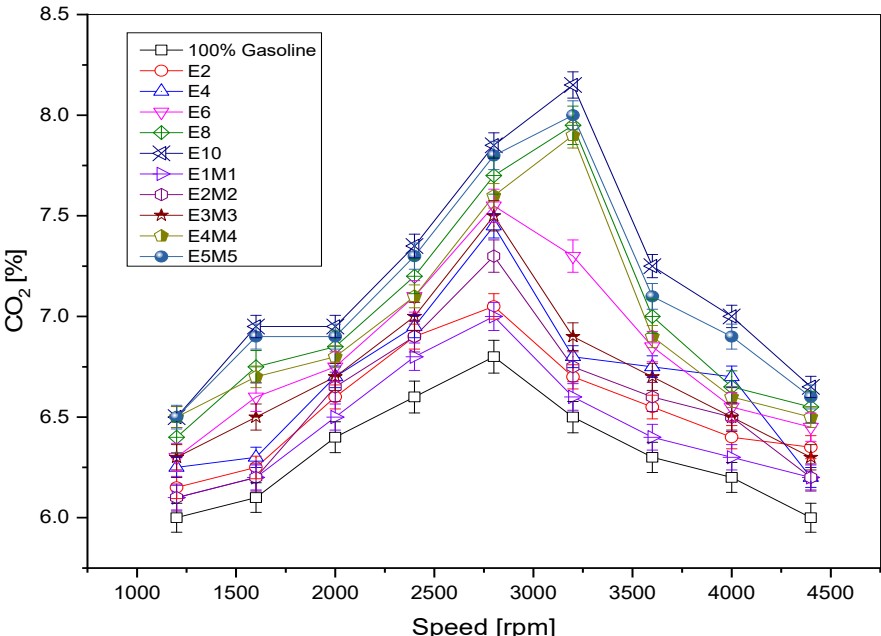

**Figure 9.** $CO_2$ emissions for the E and EM blends.

Figure 10 depicts the comparison among $NO_x$ emissions for the different blends of E and EM with gasoline. $NO_x$ emissions were noticed to be the highest for E, then EM, and the lowest for pure gasoline. E10, E5M5 and pure gasoline produced 2072 ppm, 2042 ppm and 1010 ppm $NO_x$ emission levels, respectively. E10 and E5M5 showed increases of approximately 22.7% and 20.8% in $NO_x$ emissions in comparison to gasoline, respectively. The higher $NO_x$ for E blends compared with gasoline is due to the higher EGT which, in-turn, is related to the higher temperature inside the cylinder [47]. $NO_x$ is higher for the EM blend compared with the pure gasoline, which can be explained by the observation that, with the increasing blend percentages, the combustion becomes gradually stoichiometric, and the temperature inside the cylinder increases, which in turn, also shows an increment

in $NO_x$ [26]. It is also evident that $NO_x$ is higher for E than EM because of the higher EGT of E compared to the EM blend [37]. The disintegration of the nitrogen molecule ($N_2$) into extremely reactive monoatomic nitrogen (N) is mainly responsible for higher $NO_x$ emissions. The fuel mixture containing oxygen (O) yields $NO_x$ emanations upon reacting with N. The combustion quality and development of exhaust emissions can be figured out with the interpretation of EGT [15]. The calorific value of alcoholic fuel is primarily responsible for the greater supply of blended fuel in the chamber. Subsequently, the EGT would be higher due to the more oxygenated fuel burning. The higher chamber temperature catalyzes the reaction between N and O, and subsequently, higher $NO_x$ is generated for alcoholic fuel blends. Tamam et al. [64] found increases of 1.7%, 4.2%, 6.7% and 5.8% in the $NO_x$ emissions for M10, E5M10, E10M10 and E15M10 fuels, respectively. The higher percentage increase in $NO_x$, as observed in the current study, might be due to higher fuel consumption and exhaust gas temperature in the case of blended fuels.

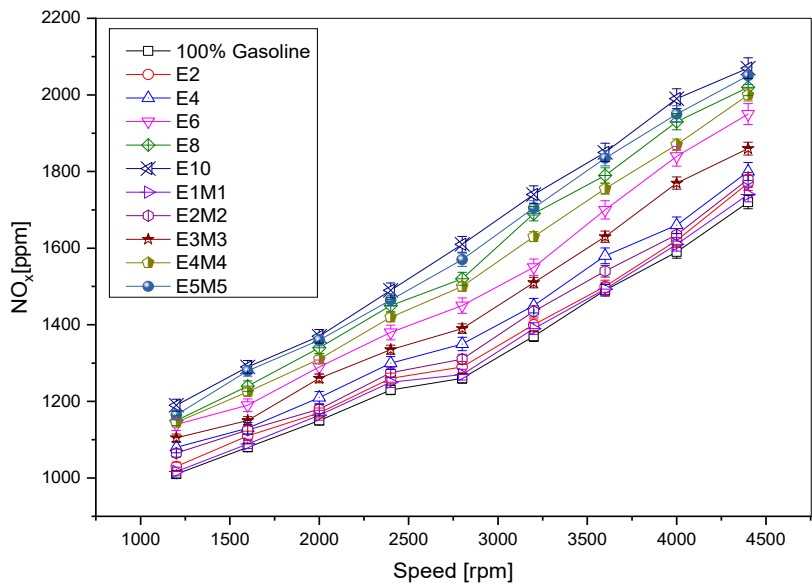

**Figure 10.** $NO_x$ emissions for the E and EM blends.

The average percentage increases in BP, BSFC, BTE, $NO_x$ and $CO_2$, and the average percentage decreases in the HC and CO emissions of E and EM blends relative to pure gasoline are given in Tables 6 and 7, respectively.

**Table 6.** Average percentage increase/decrease ($\uparrow$/$\downarrow$) in parameters for E blends.

| Fuel | Average Percentage Change in Performance Parameters and Exhaust Emissions | | | | | | | |
|---|---|---|---|---|---|---|---|---|
| E% | $\uparrow$BP | $\uparrow$BSFC | $\uparrow$BTE | $\uparrow$EGT | $\downarrow$HC | $\downarrow$CO | $\uparrow$CO$_2$ | $\uparrow$NO$_x$ |
| 2 | 1.76 | 0.06 | 0.82 | 1.64 | 2.01 | 8.98 | 3.60 | 2.10 |
| 4 | 3.61 | 0.18 | 1.56 | 7.52 | 4.27 | 4.53 | 5.62 | 5.55 |
| 6 | 4.50 | 0.92 | 1.83 | 10.3 | 6.65 | 22.3 | 7.99 | 13.4 |
| 8 | 5.08 | 1.60 | 1.99 | 12.7 | 10.7 | 30.4 | 10.8 | 18.7 |
| 10 | 5.70 | 1.97 | 2.35 | 15.4 | 13.2 | 36.2 | 13.6 | 22.7 |

**Table 7.** Average percentage increase/decrease (↑/↓) in parameters for EM blends.

| Fuel | Average Percentage Change in Performance Parameters and Exhaust Emissions | | | | | | | |
|---|---|---|---|---|---|---|---|---|
| E%M% | ↑BP | ↑BSFC | ↑BTE | ↑EGT | ↓HC | ↓CO | ↑CO$_2$ | ↑NO$_x$ |
| 1 | 0.79 | 1.02 | 0.07 | 0.77 | 0.88 | 4.12 | 2.11 | 1.03 |
| 2 | 2.04 | 1.56 | 0.58 | 4.41 | 3.22 | 8.75 | 4.04 | 3.74 |
| 3 | 3.48 | 2.49 | 0.78 | 9.05 | 4.45 | 18.5 | 6.15 | 9.33 |
| 4 | 4.19 | 3.43 | 0.80 | 11.6 | 8.63 | 26.7 | 10.1 | 16.4 |
| 5 | 4.91 | 3.67 | 1.53 | 14.2 | 12.1 | 32.8 | 12.5 | 20.8 |

*3.3. Lubricant Oil Deterioration*

The majority of research efforts have been focused on the examination of the impact of alternative fuels on the environmental emissions and performance of an engine, however, rare efforts have sought to examine the impacts of the ethanol–gasoline fuel blend (E) and ethanol–methanol–gasoline fuel blend (EM) on the physicochemical properties of lubricant oil. The engine operational time as well as the ambient and inner temperature of the engine greatly impact the performance of the lubricant oil in the engine. The current study includes the examination on the effect of ethanol and EM fuel blend on the lubricant oil properties. The comparative analysis was then made with reference to the change in lubricant oil characteristics for 120 h of an engine running on both ethanol and subsequently a methanol–ethanol–gasoline fuel blend. The change in the physicochemical characteristics of the lubricant oil operating on E10, E5M5 and gasoline (G) in comparison with fresh sample of lubricant oil is displayed in Figure 11. The kinematic viscosities of the lubricant oil at 100 °C and 40 °C were ascertained in accordance with ASTM D445. The (KV)$_{40°C}$ of the deteriorated lubricant oil for gasoline, E10 and E5M5 decreased by 11.45%, 9.64% and 6.87%, respectively, as compared to the fresh lubricant sample. Likewise, the (KV)$_{100°C}$ of the deteriorated lubricant oil for gasoline, E10 and E5M5 was decreased by 18.47%, 15.29% and 11.47%, respectively. The decline in the KV of the degraded lubricant oil was mainly due to fuel dilution and bond breakage between the lubricant oil molecules [65]. The decreases in (KV)$_{100°C}$ and (KV)$_{40°C}$ in the case of E5M5 were 3.82% and 2.77% lower than E10. The KV decreased less in the case of E5M5 due to the higher density of methanol. Moreover, the lower exhaust gas temperature in the case of E5M5 ensured less bond breakage between the lubricant oil molecules compared to E10. The decrease in the viscosity index of the lubricant oil is a primary indication of deterioration. The viscosity index (VI) of the lubricant oil was ascertained in accordance with the ASTM D2270 standard. The degradation rate was found to be lower for E10 than gasoline and E5M5 because of a 2.52% smaller decrease in the viscosity index of the lubricant oil in the case of E10. The main reasons for the greater decline in VI are more acidic sludges, contamination and oxidation [65]. The ASTM D4739 standard was followed to determine the total base number (TBN) of lubricant oil. The total base number (TBN) determines the alkaline nature of lubricant oil. It is directly proportional to the resistance in the formation of acid in the lubricant oil. The formation of acid is mainly responsible for oxidation and corrosion, but the alkaline nature of lubricant oil plays its role in the neutralization of acids. The decline in TBN can be observed for all the fuels in Figure 10. The average decline in TBN was 12.24%, 19.39% and 22.45% for gasoline, E10 and E5M5 fuels, respectively. This decline can be credited to the oxygen availability and high combustion temperatures for fusel oil, which promotes the formation of acids [66]. Moreover, the acidic nature of methanol is mainly responsible for the higher decrease in the TBN value than in gasoline and E10. Flash point is the least temperature above which lubricant oil vapors are immediately ignited by the ignition source. This is also used to identify the contamination, fuel dilution and thermal cracking of lubricant oil. The flash point (FP) of the lubricant oil was determined as per the ASTM D92 standard. The flash points for G, E10 and E5M5 decrease by 8.64%, 5.91% and 4.09%, respectively.

FP ascertains the maximum operational limits of lubricant oil, which depicts fire resistivity during operations. The lower flash point signifies a danger during lubricant oil operation

and ultimately results in system malfunction. The least decrease in flash point for E5M5 was mainly due to the higher LHV of methanol. The inspection of changes in the lubricant oil properties owing to EM blended fuels will open new avenues to explore such additives and an optimum composition for a prolonged deterioration rate. However, such a material and coatings need to be developed, which will be proven helpful in reducing wear and tear.

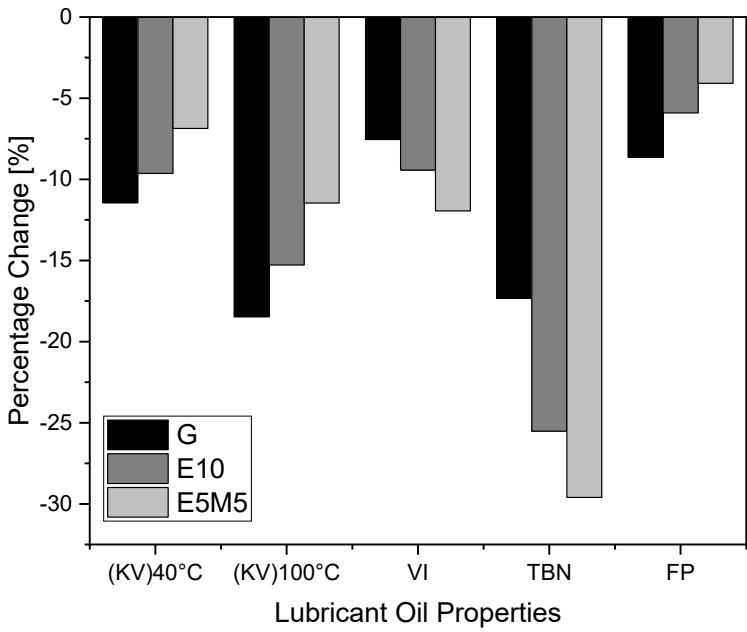

**Figure 11.** Change in the percentages of physicochemical attributes of lubricant oil.

## 4. Conclusions

The following conclusions can be deduced by experimenting the different blend percentages of E and EM in gasoline for the SI engine:

I.  Brake power increased with the increase in ethanol and methanol blends in gasoline. It was improved by 5.7% and 4.91% for E10 and E5M5, respectively, compared to pure gasoline.

II.  BSFC noted the minimum for pure gasoline compared to E and EM blends. Amongst all blends, EM showed the greater BSFC. It increased by 3.67% and 1.97% for E5M5 and E10, respectively, compared to pure gasoline.

III.  BTE increased with the increasing ethanol and methanol blending compared to pure gasoline. It can be accredited to a higher oxygen content and octane rating in alcoholic fuels. E blends showed a higher BTE competed with EM blends. BTE for E10 and E5M5 increased by 2.35% and 1.53%, respectively, in comparison with pure gasoline.

IV.  HC emissions decreased by increasing ethanol and methanol blends compared with pure gasoline. E blends showed lower HC emissions than EM blends. E10 and E5M5 showed 13.2% and 12.1% less HC emissions compared with pure gasoline.

V.  Ethanol and methanol blending decreased the CO emissions. Reductions of 36.2% and 32.8% were observed for E10 and E5M5, respectively. However, $CO_2$ emissions increased by 13.6% and 12.5% for E10 and E5M5, respectively.

VI.  $NO_x$ emissions increased for ethanol and methanol blends due to enhanced EGTs. E10 and E5M5 emitted 22.7% and 20.8% more $NO_x$ relative to pure gasoline, respectively. However, E5M5 produced 1.9% lower $NO_x$ emissions than the E10 blend.

VII.  In terms of sustainability, both methanol and ethanol have their advantages and disadvantages. Ethanol is a better option in terms of reducing greenhouse gas emissions and air pollution, as it has a lower carbon footprint compared to methanol. However, the production of ethanol can have negative impacts on land use, water resources and food security if not properly managed. On the contrary, methanol possesses a higher

energy content and can be more efficiently produced, but it has a higher toxicity level and can be harmful to human health if not handled properly.

The final outcome of this study is that the E10 fuel blend performs better than the E5M5 fuel blend in terms of higher brake power and brake thermal efficiency at the cost of lower fuel consumption, as well as lower HC and CO emissions. However, $CO_2$ and $NO_x$ emissions were higher for the E10 blend as compared to the E5M5 blend. Therefore, the ethanol–methanol gasoline blend could be used to lower the $CO_2$ and $NO_x$ emissions. The lubricant oil deterioration was prolonged for the E10 fuel blend in comparison to the E5M5 fuel blend. The percentage change in the physicochemical properties of lubricant oil was less for the E10 fuel blend. The smaller decline in the TBN and VI of lubricant oil in the case of E10 indicates its higher stability against aging and deterioration in comparison with E5M5. In addition, the current work may be augmented by post-treatment technology (reduction type catalytic converter) for a sustainable environment. Moreover, a higher combustion temperature for alcoholic fuels is responsible for the early degradation of lubricant oil, which greatly affects the engine performance. In the future, such additives need to be developed for lubricant oil which may prolong the deterioration rate, even when alcoholic fuel blends are used.

**Author Contributions:** Conceptualization, M.U. and M.A.I.M.; methodology, T.N.C.; formal analysis, F.R., M.M.A. and M.A.I.M.; investigation, S.R. and M.A.; resources, M.M.A. and M.A.K.; data curation, F.A.M. and H.M.A.; writing—original draft preparation, M.A.I.M. and Y.F.; writing—review and editing, F.R., M.M.A. and M.U.; supervision, M.U. All authors have read and agreed to the published version of the manuscript.

**Funding:** This research was funded by the Researchers Supporting Project number (RSPD2023R698), King Saud University, Riyadh, Saudi Arabia. This work has been partially funded by Abu Dhabi University, UAE.

**Institutional Review Board Statement:** Not applicable.

**Informed Consent Statement:** Not applicable.

**Data Availability Statement:** Not applicable.

**Acknowledgments:** The authors extend their appreciation to the Reserachers Supporting Project number (RSPD2023R698), King Saud University, Riyadh, Saudi Arabia for funding this research work.

**Conflicts of Interest:** The authors declare no conflict of interest.

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
