# Peer review of "Comparative Assessment of Ethanol and Methanol–Ethanol Blends with Gasoline in SI Engine for Sustainable Development"

_sustainability, doi:10.3390/su15097601_

Round 1

Reviewer 1 Report

The data provided by the authors are valuable and very interesting, although the manuscript has just small issue that should be revise carefully:

1.     Some new and recent studies should be added to the introduction section to address the latest findings in the field, especially situations related to toxic emissions with their impacts on human health and environment. Also, there is no in-depth analysis for published works related to different nanoparticles in diesel engines. Authors should fill those gaps with 10.1016/j.fuel.2022.126377 and 10.1016/j.apr.2021.101305. 

Author Response

We would like to thank Reviewer 1 for encouraging and suggesting valuable papers which will definitely improve the quality of the literature review. However, more relevant articles have been added in the introduction section of the revised manuscript.

Reviewer 2 Report

Dear Authors,

-      The manuscript deals with ethanol and ethanol-methanol blended gasoline, from which ethanol blended part is a widely researched area; therefore, the effect of methanol should be highlighted.

-      It needs much improvement to be scientifically correct.

-      The lines need to be numbered, so reviewing is much more difficult.

-      Major and minor changes need to be made. 

A detailed review have been attached.

Author Response

Rebuttal File is attached.

Reviewer 3 Report

Comments:

1. Can you add brief information about your work and outcomes in Introduction section?

2. Write the information about chemicals? Did you miss the Materials and method section?

3. Explain about what is SI engine and why did you select it for this study? 

4. What is the final conclusion of your study? which gasoline mixture is better to use?

4. Please refer the attached document for more comments.

Author Response

Reviewer 3

We appreciate and thank the reviewer for the valuable suggestions made.

1.      Can you add brief information about your work and outcomes in Introduction section?

Brief information regarding the research approach and achieved outcomes has been added in the end of the Introduction section of the revised manuscript.

2.      Write the information about chemicals? Did you miss the Materials and method section?

The materials and method section (Section 2) has been added in the revised manuscript to have a clear understanding between the chemicals used and the test approach.

3.      Explain about what is SI engine and why did you select it for this study?

A spark ignition (SI) engine is a type of internal combustion engine that uses a spark plug to ignite the fuel-air mixture in the combustion chamber at end of compression stroke. Therefore, the air-fuel mixture undergoes combustion and expanding gases cause the piston to move downward and turn the crankshaft in order to rotate wheels of vehicle. SI engine use gasoline as fuel. A SI engine has been selected for the current study due to its multiple advantage over diesel engine like lower emissions, light weight, less noise and vibration and fuel efficiency.

4.      What is the final conclusion of your study? which gasoline mixture is better to use?

The final outcome of the study is that the E10 fuel blend perform better than E5M5 fuel blend in terms of higher brake power and brake thermal efficiency at the cost of lower fuel consumption, HC and CO emissions. However, CO2 and NOx emissions were higher E10 blend as compared to E5M5 blend. Therefore, the ethanol-methanol-gasoline blend could be used to lower CO2 and NOx emissions.

5.      Please refer the attached document for more comments.

The comments mentioned in the attached document have been addressed in the revised manuscript.

Round 2

Reviewer 2 Report

Dear Authors,

According to the reviewer’s opinion, the manuscript has been significantly improved.

However, let me make a few more comments:

1. Double-check the spelling throughout the text.

(i) A space should be inserted in several places: e.g. B12(12% butanol in 88% gasoline); 1500rpm; Soudagar et al.[23]; …

(ii) Correct spelling of the chemical molecule: NOx, CO2, etc., where x and 2 are in subscripts.

(iii) % as unit of measure: 6 and 5.5%; 5 and 2%; 31 and 14%; 35 and 15%. It would be necessary everywhere.

(iv) font sizes change continuously in the text.

2. Section: Test setup

(i) It is essential to show how the fuel mass is formed from the volume of the burette for the BSFC.

(ii) It would be helpful to describe the measuring principles of the gas analyzer.

3. Diagrams in results: Speed [rpm] would be recommended.

4. Section: Lubricant oil deterioration

(i) all materials, setups and methods that belong here should be placed in the primary Materials and Methods chapter, because this is the Section for Results. ASTM standards should be referenced.

Author Response

  1. Double-check the spelling throughout the text.

(i) A space should be inserted in several places: e.g., B12 (12 % butanol in 88 % gasoline); 1500 rpm; Soudagar et al. [23]; …

Thanks for suggesting a correction. A space has been inserted in the mentioned places as well as at appropriate places in the whole manuscript.

(ii) Correct spelling of the chemical molecule: NOx, CO2, etc., where x and 2 are in subscripts.

Thanks. Changes are incorporated in revised version.

(iii) % as unit of measure: 6 and 5.5%; 5 and 2%; 31 and 14%; 35 and 15%. It would be necessary everywhere.

Thanks for the comment. % as a unit of measure has been added with each value.

(iv) font sizes change continuously in the text.

Font size has been made uniform as per the template of the Sustainability Journal.

  1. Section: Test setup

  • It is essential to show how the fuel mass is formed from the volume of the burette for the BSFC.

Thanks for your suggestion. The densities of gasoline, methanol and ethanol are 0.736, 0.791 and 0.785 g/ml. The mass is calculated from the formula which is density= mass/volume. The mass of gasoline, methanol and ethanol for 30ml fuel supply are 22.08, 23.73 and 23.55 g. The mass of gasoline, methanol and ethanol for 50ml fuel supply are 36.8, 39.55 and 39.25 g. The mass of gasoline, methanol and ethanol for 100ml fuel supply are 73.6, 79.1 and 78.5 g.

(ii) It would be helpful to describe the measuring principles of the gas analyzer.

Thanks for the suggestion. EMS 5002 relies on non-dispersive infrared (NDIR) sensors that emit infrared radiation at specified wavelengths in case of CO2 and CO emissions. These wavelengths pass through the sample tube being analyzed and a reference tube that contains a non-absorbing gas. The wavelengths are measured based on the absorption rate of the gases. The electrochemical sensor used to sense NOx emissions through electrical signals generated as a result of chemical reaction between nitrogen and air. HC emissions are identified by flame ionization detection inside the emission analyzer, as electrons or ions produced when hydrocarbon burned in a hydrogen flame. These ions and electrons can be measured by the collector electrode placed near the flame.

  1. Diagrams in results: Speed [rpm] would be recommended.

Thanks. Diagrams in results have been modified with the x-axis as Speed [rpm].

  1. Section: Lubricant oil deterioration

  • all materials, setups and methods that belong here should be placed in the primary Materials and Methods chapter, because this is the Section for Results. ASTM standards should be referenced.

Thanks for the suggestion. The description regarding lubricant oil has been added in the Materials and Methods section along with Table 1 displaying the lubricant oil characteristics, ASTM standards, and Testing equipment in the revised manuscript.

Reviewer 3 Report

Thank you for addressing the comments to improve the manuscript.

Minor Comments: 

1. Designed methods need to be improve with the strong evidences to prove your work.

2. Need to improve the conclusion to support your entire work.

Author Response

  1. Designed methods need to be improved with the strong evidences to prove your work.

Thanks for your suggestion. Figure 1 has been added as evidence of current work done in the Thermal Power Systems Laboratory of University of Engineering and Technology, Lahore.

  1. Need to improve the conclusion to support your entire work.

Thanks for your comment. The final outcome of the study is that the E10 fuel blend perform better than E5M5 fuel blend in terms of higher brake power and brake thermal efficiency at the cost of lower fuel consumption, HC and CO emissions. However, CO2 and NOx emissions were higher E10 blend as compared to E5M5 blend. Therefore, the ethanol-methanol-gasoline blend could be used to lower CO2 and NOx emissions. The lubricant oil deterioration was prolonged for E10 fuel blend in comparison to E5M5 fuel blend. The percentage change in physicochemical properties of lubricant oil was less for E10 fuel blend. The less decline in TBN and VI of lubricant oil in case of E10 indicates its higher stability against aging and deterioration in comparison with E5M5.